# Scaling up of Eco-Bio-Social Strategy to Control *Aedes aegypti* in Highly Vulnerable Areas in Fortaleza, Brazil: A Cluster, Non-Randomized Controlled Trial Protocol

**DOI:** 10.3390/ijerph18031278

**Published:** 2021-01-31

**Authors:** Suyanne Freire de Macêdo, Kellyanne Abreu Silva, Renata Borges de Vasconcelos, Izautina Vasconcelos de Sousa, Lyvia Patrícia Soares Mesquita, Roberta Duarte Maia Barakat, Hélida Melo Conrado Fernandes, Ana Carolina Melo Queiroz, Gerarlene Ponte Guimarães Santos, Valter Cordeiro Barbosa Filho, Gabriel Carrasquilla, Andrea Caprara, José Wellington de Oliveira Lima

**Affiliations:** 1Collective Health Postgraduate Program, State University of Ceará, Fortaleza 60714-903, Brazil; kellyanneabreu@gmail.com (K.A.S.); renatinhaam28@gmail.com (R.B.d.V.); izasousa222@gmail.com (I.V.d.S.); lyvia_mesquita@hotmail.com (L.P.S.M.); robertadumaia@gmail.com (R.D.M.B.); helidamelopsi@gmail.com (H.M.C.F.); acmq28@hotmail.com (A.C.M.Q.); gerarlenepg@hotmail.com (G.P.G.S.); valtercbf@gmail.com (V.C.B.F.); andreacaprara1@gmail.com (A.C.); jwolima@yahoo.com.br (J.W.d.O.L.); 2Nursing Department, Federal University of Piauí, Picos 64607-670, Brazil; 3Federal Institute of Education, Science and Technology of Ceará, Aracati Campus, Aracati 62800-000, Brazil; 4Santa Fe de Bogotá Foundation, Bogota 110311, Colombia; gcarrasquillag@gmail.com

**Keywords:** health impact assessment methods, *Aedes aegypti*, Brazil, scaling up, community-based intervention, dengue, chikungunya, Zika virus, mixed methods, vector control

## Abstract

*Aedes aegypti* is a cosmopolitan vector for arboviruses dengue, Zika and chikungunya, disseminated in all Brazilian states. The Eco-Bio-Social (EBS) strategy is vital in *Aedes aegypti* control as it mobilizes stakeholders (government, professionals, society, and academics) to promote healthy environments. This paper describes the rationale and methods of expanding the EBS strategy for *Aedes aegypti* control in Fortaleza, Northeast Brazil. A cluster, non-randomized controlled clinical trial was developed to analyze the strategy’s effectiveness in vulnerable territories (high incidence of dengue and violent deaths; low HDI; substandard urban infrastructure, high population density, and water scarcity). We selected two intervention and two control groups, resulting in a sample of approximately 16,000 properties. The intervention consisted of environmental management by sealing large elevated water tanks, introduction of beta fish in waterholes, elimination of potential breeding sites, and mobilization and training of schoolchildren, endemic disease workers, health workers, social mobilizers, and community leaders; community surveillance of arboviruses; construction and validation of a booklet for the prevention of arboviruses in pregnant women. We analyzed the costs of arboviruses to government and households, the intervention cost-effectiveness, chikungunya’s chronicity, and acceptance, sustainability, and governance of vector control actions. The primary outcome (infestation) was analyzed using the house, container, and Breteau indices. We hope that this study will help us understand how to scale up strategies to fight *Aedes aegypti* in vulnerable areas.

## 1. Introduction

Millions of people get sick and die from diseases transmitted by *Aedes aegypti*. For example, a study on dengue’s public health burden in 2010 estimated about 390 million Dengue virus (DENV) infections (95% range: 284–528) annually, of which 96 million (67–136) manifest themselves. Asia bears 70% of these infections, Africa 16%, the Americas 14%, and Oceania’s countries less than 0.2% [1]. In Brazil, the number of Dengue cases hiked 232.7%, up from 790,834 in 2000 to 2,631,767 cases in 2015. The rates of Years of Life Lost (YLL), Years Lived with Disability (YLD) and Disability-Adjusted Life Years (DALY) increased by 420.0%, 187.2%, and 266.1%, respectively [2]. A total of 11,137,664 probable dengue cases were reported in Brazil from 2003 to May 2019. Five epidemic years occurred during this period: 2008, with the circulation of DENV-2; 2010 and 2013, with the reintroduction of serotypes; 2015 and 2016, after the entry of two new arbovirus, chikungunya virus (CHIKV), and Zika virus (ZIKV) [3].

Cities with rapid population growth, disorderly use of space and housing construction, and with water supply and solid waste collection restrictions provide ideal conditions for *Aedes aegypti*, which is highly adapted to the urban environment and anthropophilic [4]. Several global regions and practically all Brazilian cities experience these issues. In Fortaleza, dengue has historically manifested itself most significantly in peripheral neighborhoods lacking social policies, structured urbanization, regular municipal water supply, satisfactory garbage collection, good housing conditions, and broad community participation in vector control actions [5]. Furthermore, these territories are marked by violence that often hampers health professionals’ activities, impairing preventive actions, and aggravating the epidemiological situation [6].

There are currently no licensed vaccines para Zika e chikungunya [7,8] and the licensed vaccine for dengue is only safe for seropositive [9]. Given the above and the impossibility of immunizing the population against all transmitted viruses, people must fight against the mosquito. However, this is a too tortuous task due to factors that involve the relationship between man, the vector, the environment, those mentioned above, high demographic density, socioeconomic inequalities, fragile social capital, inappropriate destination of river waters and wastewater, resistance to the use of insecticides, urban mobility, and insufficient health services to respond to the demands of the population [10,11,12,13].

Also, systematic reviews and meta-analysis evidence knowledge gaps in the effectiveness of vector control techniques [14,15]. A meta-analysis analyzed publications on environmental control actions (tank sealing with materials impregnated or not with insecticides, waste management with and without direct garbage collection, and elimination of breeding sites). It concluded that high-quality studies are required to strengthen evidence of an actual decline in entomological indices. We also observed that most low-quality studies addressed sealing tanks with lids without insecticides [16].

Research using the EBS strategy fills these gaps because it consists of a package with low-cost actions such as sealing large tanks and eliminating or managing potential breeding sites, planned and carried out with the collaboration of the interested parties, that is, the community, government, and scholars [17,18,19,20,21,22]. By eliminating the most productive containers, the EBS strategy reduces vector density and, consequently, the number of reported dengue cases [20]. Figure 1 presents the strategic principles and areas of operation of the EBS strategy [20].

Successful results were achieved from 2011 to 2013 using a package of interventions, including controlling potential breeding sites, installing window nets with insecticide-treated materials, coverage of the main water tanks, and partnership between vector control services, community, and stakeholders. This Cluster Randomized Controlled Trial was developed in Brazil (Fortaleza), Colombia (Girardot), Mexico (Acapulco), Ecuador (Machala), and Uruguay (Salto), and each country adapted the interventions according to the participation of the community involved [23].

In Fortaleza, a Cluster Randomized Controlled Trial comparing intervention groups was carried out with ten control groups. Each group consisted of 100 households. The intervention included community workshops, community sensitization for home and peridomicile cleaning, coverage of elevated containers, mobilization of schoolchildren and older adults, and distribution of information, education, and communication materials. The results pointed to the intervention’s effectiveness, compared to the control consisting of the government’s routine actions [24].

In response to the growing dengue and the introduction of chikungunya and Zika, in 2017, Canada’s International Development Research Center (IDRC) financed the expansion of these studies in three Latin American countries (Colombia, Mexico, and Brazil). Concerning the Brazilian study, a Cluster Randomized Controlled Trial would be the most appropriate design due to the high level of evidence, avoiding systematic errors, striving for impartiality [25], and because the phenomenon studied (*Aedes aegypti* infestation) is intrinsically related to the territory. However, it was decided to test the EBS strategy’s effectiveness in high epidemiological risk and social vulnerability territories, which were located in the southern suburbs of Fortaleza [26]. Therefore, the selection of households was intentional. The aim of the study was to advance knowledge on reducing *Aedes aegypti* infestation rates in challenging vector control territories and foster positive changes on a large scale to promote health and prevent social and economic spoliation caused by dengue, Zika, and chikungunya in vulnerable communities. The territory’s epidemiological and social context made the process challenging and innovative, which increased knowledge and led to the construction of lightweight technologies such as educational and managerial tools [18].

The general research question was: How effective is the EBS strategy on a large-scale trial, comparing EBS strategy plus vector control routine with solely vector control routine, in households with a history of high dengue incidence and strong social vulnerability?

Specifically, we aimed to describe the costs of the three arboviral infections to the government and the households and evaluate the intervention’s cost-effectiveness; to analyze intersectionality and effects of chronic chikungunya in women; to analyze acceptance and sustainability of sealing large tanks, governance of actions to combat arboviruses, and the impact of environmental management on entomological indices; to provide training in new vector control technologies for field workers; to implement participatory surveillance for early detection of dengue, chikungunya, and Zika cases; and to develop knowledge management on arboviruses for participatory action in school settings.

We hypothesized that scaling up the EBS strategy with community mobilization for environmental management effectively reduces the density of *Aedes aegypti* in clusters of high social vulnerability and epidemiological risk. In settings where small containers and large elevated and open water tanks prevail, eliminating the condition of potential breeding of small containers and the sealing of large tanks are inexpensive and effective actions in reducing *Aedes aegypti* infestation, especially in territories with population density and housing and financial infrastructure limitations.

Mobilization targeting different actors to enhance the resources and social network available in the community can expand the intervention’s scope at the individual, group, and cluster level, by strengthening the leading role of the population in vector control. Therefore, it is admittedly effective in mobilizing community groups through dialogue and promotion of training to combat mosquitoes and foster surveillance of suspected cases of arboviruses. In this sense, schoolchildren and other groups, such as older adults engaging in physical exercises and community leaders, are crucial individuals in the community mobilization process.

## 2. Materials and Methods

### 2.1. Trial Design

This is a scaling-up, cluster randomized controlled trial designed to analyze the effectiveness of the EBS strategy, compared to the usual control of *Aedes aegypti* [24], which aimed to expand the first study in three avenues: (1) Increasing the number of participating households; (2) Expanding the intervention’s benefits; and (3) Expanding the knowledge about territories with high epidemiological risk for dengue and social vulnerability.

We analyzed the impact of these actions on the frequency of household breeding sites for *Aedes aegypti*, and household *Aedes aegypti* infestation, evaluated in three stages, in the intervention and control households. The first occurred before the actions, the second during the actions, and the third after the actions.

The community’s mobilization activities lasted 24 months. This study was registered in the Brazilian Registry of Clinical Trials (Register Number: RBR-6ck7g2) under the title “EXTENDED ACTIONS TO PREVENT AEDES AEGYPTI-TRANSMITTED DISEASES” (Free translation from Portuguese), available at: http://www.ensaiosclinicos.gov.br/rg/RBR-6ck7g2/. We followed the Consort 2010 statement’s recommendations: extension to cluster randomized trials [27].

### 2.2. Study Setting

Fortaleza is the fifth most populous capital of Brazil, with an estimated 2,669,342 inhabitants for 2019. In 2010, it had a demographic density of 7786.44 inhabitants/km^2^; 74% of households with adequate sanitation; 74.8% of urban households on wooded public roads and 13.2% with adequate urbanization (manholes, sidewalks, pavements, and curbs) [28]. Besides the high population density and uncoordinated urbanization, tropical climate, co-circulation of serotypes of the DENV, the CHIKV, and the ZIKV, contribute to an annual seasonal pattern of illness and cycles of epidemic and interepidemic years [26].

Arboviruses occur heterogeneously in the 121 neighborhoods and six health regions of Fortaleza. In the 2011–2015 period, dengue’s burden was more significant in the southern suburbs and strongly associated with violence, poverty, and irregular solid waste collection [26]. In this context, the Fortaleza suburban area is a vital scenario for testing a large-scale health strategy.

### 2.3. Eligibility Criteria

All families domiciled in the selected households could participate in the study. People who were not in the territory at the time of the interventions were excluded from that sample but could participate if they so wished in a later collection.

In schools, the activities were directed to students over the age of six enrolled in elementary school. We chose this age group because we believe that children of six years of age and over would understand and participate in the research team’s activities. Schoolchildren who did not present an Informed Consent Form (ICF) signed by the parents were excluded from the sample.

### 2.4. Sample Size

Cluster size was defined, aiming at the largest possible expansion of the sample given the available resources [24]. Thus, each cluster consisted of 4000 properties, totaling a sample of 16,000.

The baseline consisted of entomological data (Entomological Survey Before—ES Before) of all properties whose responsible people consented to receive Endemic Disease Control Agents (EDCA). The remaining entomological surveys occurred in 50% of the properties selected by a draw. These quantitative data measured the primary outcome (infestation). The costing study measured the direct and indirect costs of dengue and chikungunya to households in the four territories (*n* = 396); the municipal costs for the prevention and treatment of dengue, Zika, and chikungunya (from 2017 to 2018); and cost-effectiveness of the EBS strategy. Qualitative data measured the scope [29] of the EBS strategy. The number of participants for each intervention activity is shown in Table 1:

Thirteen people were interviewed in the chikungunya chronicity study. Five managers were interviewed in the governance study, and 23 households in the sustainability study.

### 2.5. Participant Schedule

Table 2 presents the summary of the research activities schedule. Three Entomological Surveys (ES) were carried out: before, during, and after the intervention. The first, ES-Before, between January and May 2018 (rainy season) and was the study’s baseline. School and community activities were planned during this period. In the intervention areas, after collecting data, EDCA seized the opportunity and immediately carried out the environmental management guidelines. The ES-During occurred from September to December 2018 (dry season) and the ES-After, from February to May 2019 (rainy season). Immediately after collecting data from the ES-Before, the large elevated tanks’ sealing was provided to the community, giving rise to the EBS strategy’s onset. EBS approach training was held from June to September 2018, activities in school settings occurred from May 2018 to November 2019, and community case surveillance was staged between February and December 2019.

Data on the cost-effectiveness of the intervention concerning municipal routine vector control and the costs of dengue, Zika, and chikungunya to the population (from the intervention and control areas) and the government were collected between August 2018 and December 2019.

In 2020, we analyzed governance in the actions to combat arboviruses to understand government officials’ current dynamics; the technical, management, and leadership skills; the power relationships, and community participation. All activities occurred as scheduled. The evaluation was, therefore, successfully concluded.

### 2.6. Feasibility Study and Allocation of Clusters

It is essential to conduct a highly complex feasibility study to raise critical issues, such as the epidemiological, ecological, and social contexts [29,30] before starting an intervention that comprises several interaction components. Initially, we selected a regional health administration in the southern suburbs of Fortaleza, Brazil, to assess the study’s feasibility. Working in a single regional health administration facilitated researchers’ travel and strengthened the bond between the community, researchers, managers, and health and education professionals.

Feasibility was tested in three stages: (1) analyzing, by neighborhood, the prevalence of dengue and *Aedes aegypti* infestation in the last five years; (2) consulting with EDCA responsible for the supervision of the neighborhoods chosen in step 1 on social and ecological aspects; (3) performing a quick survey to learn about *Aedes* potential breeding sites.

**Step one**: the researchers met with the officials responsible for Fortaleza’s epidemiological surveillance to assess the epidemiological context [29], analyzing data on dengue cases and *Aedes* infestation in the last five years. Four neighborhoods with a similar epidemiological history were selected based on this information.

**Step two**: meetings were held with the endemic districts’ coordinators, EDCA responsible for the neighborhoods’ supervision to assess the ecological and social context [29] in order to choose the clusters. These professionals had more than ten years of experience in these territories, and with the help of maps, established groups consisting of approximately 4000 properties, 10,000 people, encompassing, preferably, a school at the center of the defined area. After meetings between professionals and researchers, they concluded that households had ecological and social similarities.

**Step three**: a quick survey was carried out on the potential *Aedes* breeding sites in 50% of the study’s properties to determine the potential scope of the intervention [29], and thirteen EDCA were trained to perform this task. A form (Appendix A) was used to record some water reservoirs’ frequency, home cleanliness, and the property’s infrastructure. As for the property’s infrastructure, the number of open eaves [31,32] (opening between the wall and the roof), and external accesses (number of openings that allowed the mosquito to pass from outside to inside properties and vice versa) was recorded. In each block, EDCAs visited the first house on the street, did not visit the second, and followed through with this alternate method.

Data were processed and analyzed at the end of the survey. From the results, the researchers concluded that the intervention was applicable to the context of the clusters. A draw was carried out among the four areas analyzed for the selection of intervention and control clusters.

### 2.7. Intervention

Community participation in vector control actions is essential for their effectiveness and sustainability [17,20,33]. The theoretical framework of Community-Based Participatory Research (CBPR) was used to guide the intervention because the partnership between researchers and stakeholders results in more meaningful and sustainable initiatives, besides improving the understanding of cultural, social, environmental, and governmental issues affecting people’s health [34] (Figure 2). Figure 3 shows the actions that were developed, the places where they were held (homes, schools, community or services), and their objective (planning, intervention, and evaluation).

Community activities started immediately after the ES-Before by sensitizing the intervention areas’ residents through the EDCAs to properly manage potential breeding sites and sealing the large elevated water tanks.

Studies show that large water reservoirs are the main breeding sites for *Aedes aegypti* [33,35,36], and sealing these devices reduces infestation. Therefore, it is a recommended technique for vector control [20,37,38]. In Fortaleza, frequent water shortage causes insecurity and the need to store water, mainly in suburban neighborhoods [6]. Due to water and financial scarcity, the reservoirs are constructed in different shapes, sizes, materials, and, in most cases, pose a real risk to public health. The inspection of large elevated tanks is challenging to the residents. Therefore, we intended to seal as many of these containers as possible, given the financial resources available and the acceptance of the community.

EDCAs have also mapped vacant lots and sensitized landowners about the importance of cleaning and preserving these spaces. During the two-stage home visits, the EDCAs reinforced the importance of protecting the environments and people from the *Aedes* mosquito during the entomological surveys.

We aimed to mobilize the largest number of children and adolescents to control *Aedes aegypti* in the intervention areas, and schools were the interaction spaces previously chosen by the researchers. Students’ activities were carried out in three municipal schools located within the intervention areas and a private school very close geographically and attended students residing in the intervention area. Third to fifth-graders were selected from the João Nunes Pinheiro municipal school. These students volunteered to participate in the research after the invitation by the researchers. This school had no elementary school level II. In the other schools, the participants were sixth to ninth-graders and were selected by a draw among those who volunteered to participate in the proposed activities. We intended to distribute the participants proportionally to each school year. However, teachers requested additional spots in some schools, which was met by the researchers.

The following activities were developed to establish a participatory surveillance routine in school environments: psychodrama-based training for students to act in vector control [39,40]; Junior Brigade (a group of schoolchildren trained to eliminate potential *Aedes* breeding sites in their schools, in partnership with EDCAs, social mobilizers, and employees of educational institutions) [41]; *Aedes* hunters (schoolchildren trained to eliminate potential *Aedes* breeding grounds around the homes, in partnership with family members and EDCAs; associated with educational activities carried out in the school environment such as mosquito cycle exhibition; case study [42]; focus group [43]; Photovoice [44] and photo exhibition.

Pregnant women were recruited at the Primary Family Health Facility (PFHF) of the intervention areas after they were invited with their partners to join three conversation wheels in each PFHF. We aimed to identify and address these people’s knowledge gaps in the prevention of arboviruses during pregnancy. Simultaneously, the researchers collected information that supported the construction of an educational booklet for the control and prevention of arboviruses during pregnancy, using the EBS strategy. Then, 11 peers validated content and appearance [45].

The EDCAs responsible for collecting entomological data from the four research territories, health workers, social mobilizers, and community leaders from the intervention areas were offered a training course in EBS strategy using participatory methodologies anchored in Paulo Freire [46]. The following tools were used: Free Word Association Test (FWAT) [47,48], case studies [42], and in-service training [49]. The practical part was only offered to EDCAs because the study design required these professionals’ consistent collaboration. After all, this content relates to their work. They do not have sufficient instructors to monitor individually all those who participated in the practical part in the field [50].

The recruitment of various community groups (religious groups, physical activity groups, community gardens, and community groups) to set up a surveillance network for arbovirus cases occurred by invitation. Those who agreed to participate were offered a three-meeting training program that addressed the mosquito’s life cycle, dengue, Zika, and chikungunya signs and symptoms, and ways of preventing arboviruses [51]. A before-and-after test instrument was applied to collect previous knowledge about *Aedes aegypti*, measures to prevent illness due to dengue, Zika, and chikungunya arboviruses and control practices, and knowledge (re) built after training.

WhatsApp group participants were registered in the end, and we advised that every suspected case of arbovirus should be reported [52,53]. When a suspected case was reported on the next business day, someone from the municipal epidemiological surveillance went to the place and performed the procedures established in the municipal health service routines [51]. Information about arboviruses and pictures of areas vulnerable to mosquito proliferation were often posted to the group to motivate and keep watchers on the network.

### 2.8. Entomological Surveys

Entomological surveys were carried out before (ES-Before), during (ES-During) and after (ES-After) the intervention, which covered two rainy seasons in the city of Fortaleza: 2018, from February to June, and 2019, from February to June.

In the ES-Before, researchers and EDCAs met to standardize the collection techniques [38] and improve the use of the data collection form (Appendix A). After three meetings, which totaled 24 h, the EDCAs were distributed among the four households, aiming to visit all the territories’ properties in question. Three visits were made per property at different times and days.

Information was collected on all properties built within the perimeter defined in the feasibility study and which the responsible person agreed with the EDCAs’ visit. The researchers registered all properties on forms, so that the next surveys, only 50% were drawn to receive the home visit. At the time, data were collected on the structure of the property and features of the inhabitants such as frequency of open eaves and other mosquito entry and exit accessways, for example, from non-netted windows and doors, the number of inhabitants, and women of childbearing age. The form also had a field for recording the household caregiver’s unique conditions, such as physical limitations, alcohol abuse, night or weekend home presence.

Regarding potential *Aedes* breeding sites, the frequency of all containers that could accumulate water from the public supply system, rain, waterholes, well, or appliances, such as refrigerators, water dispensers, and air conditioning equipment, was recorded. The form contained specific fields for recording the frequency of small containers (reservoirs capable of accumulating up to 100 L of water) and the frequency, material (metal and plastic), shading, and sealing conditions of large containers (capacity exceeding 100 L of water). For instance, small buckets, bottles, tires, large storage tanks, water tanks, tanks, and pools were considered. The collection of more information about the large storage tanks is because they can promote habitat for greater reproduction of *Aedes* and, generally, they remain in the property for a longer time requiring management for an extended period [6,19] (Figure 4).

In this study’s entomological surveys, time to search for the immature forms of *A. aegypti* in the various potential breeding sites of a residence ranged from 10 to 60 min, depending mainly on the number of large storage tanks the household and the difficulty accessing them. So, in each large storage tank, the search for immature forms lasted longer when there were no immature forms, or it was interrupted when 10 to 15 immature forms of culicids were retrieved. These collected specimens were then sent to a laboratory, where they were also classified up to species [54,55]. In every entomological survey, all types of water containers, located indoors and outdoors, of all selected households, were searched for the presence of immatures forms of *Aedes aegypti*.

The frequencies of small and large storage tanks were recorded, and a larval collection was performed in ES-During and ES-After. The properties with sealed large tanks not included in the draw were also visited during the last data collection to verify the intervention’s sustainability. In the last collection, the frequency of water tanks, tanks, drums, cisterns, waterholes, gutters, drains, and buckets was also recorded, which, if infested, allowed the mosquito immediate access to residents, hence the use of nets on doors and windows as a vector control measure.

Ovitraps were used to monitor *Aedes* infestation in December 2018 and January 2019. It was a period of low infestation according to the entomological indices recorded by larval collection. Thus, we decided to use ovitraps as the most sensitive method in low infestation scenarios [56]. Two properties were chosen in each block to install the traps, one with infestation and the other without infestation in the ES-During. A trap was positioned inside the property and another one outside. The external ovitrap was positioned in a shaded place and protected from the rain.

Colonial grass syrup (*Panicum maximum*) was produced with 42 g of grass, fermented for 15 days in a container with 5 L of water. It was then diluted to 10% in mineral water, and 300 mL were applied to each trap [57]. The ovitraps consisted of plastic vases, black, matte, with wire handle and cement base for the vertical fitting of a wooden rectangle (palette) (10 × 3 cm^2^), with a smooth and porous side, which was close to the container wall.

The traps were repositioned weekly. During surveys, the palettes were collected, stored in wooden crates, and sent to the laboratory, where the eggs were counted and classified up to the genus using a stereoscopic microscope (20×).

The following indices were analyzed: (a) Ovitrap Positivity Index (OPI) = (No. of positive traps/No. of ovitraps inspected) × 100; (b) Egg Density Index (EDI) = (Total eggs in the palettes/Total positive traps); and (c) Mean Egg Index (MEI) = (No. of eggs collected/No. of ovitraps inspected [54].

### 2.9. Sealing of Water Tanks

The masonry tanks were covered with nylon mesh, consisting of interwoven fibers with a 1 mm opening, fixed with nails and plaster or cement. Fiber, plastic, or asbestos tanks were fixed with galvanized wire N° 18. It is noteworthy that the professionals who performed the intervention were construction workers, trained and supervised by people experienced in using the nylon mesh sealing technique.

In one cluster, we managed to seal all open water tanks, whose owners authorized by signing the informed consent form. More storage tanks were found in another, but it was impossible to provide sealing to all households. However, the materials, guidance, and supervision were offered if the person in charge of the property provided someone to cover their large elevated tanks.

### 2.10. EBS Strategy Training with EDCAs, Community Health Workers (CHW), Social Mobilizers, Health Surveillance Workers (HJW), and Community Leaders

EDCAs, CHW, social mobilizers, HJW, and community leaders are crucial for implementing an EBS strategy, as they are in frequent contact with the community and have their trust. Our experience in high social vulnerability areas pointed out that it is highly recommended to establish the first contact with these leaders before immersing in the territory [50].

Therefore, a course was promoted to present the EBS strategy in-depth, strengthening the partnership between these stakeholders and researchers and encouraging professionals to use the EBS strategy principles in their daily activities. The agreement with the participation terms was documented by signing the informed consent form.

The training course was designed and elaborated from an educational guide [50], a teaching material intended for educators to bring them closer to the EBS approach and the emancipatory teaching-learning methodologies, assisting them in the foundation of dialogic moments facilitated by the formative meetings.

The guide was essentially based on the teachings of Paulo Freire [46], who advocates education as a liberating practice that breaks with any form of domination, prejudice, and violence pervading people’s social and political life. Thus, the guide emphasizes the promotion of educational practices committed to transforming real-life and work spaces and the recognition and appreciation of people and health workers, especially those acting as health watchers.

Its matrix was structured in five face-to-face theory-based meetings, on consecutive days, and field monitoring activities, lasting 13 days. The meetings corresponded to a workload of 20 theoretical hours, and the practical field monitoring, 104 h [50].

The FWAT was applied at the training course’s opening and the end of theoretical explanations. The FWAT allows people to express words that represent the common knowledge of a given group based on stimuli [47].

We used group activities with the application of case studies because it is a methodology that debates life, work, and social aspects from real situations, leading the participant to build knowledge from the reflections on the problem case [42,58]. The training led the participants to discuss real and daily work situations to transform previous knowledge through existing knowledge valorization. A culture circle was held at the close, which allowed the critical and reflective exercise of the real social problems permeating the students’ lives, enabling a democratic construction of autonomy [59].

Field activity started following the theory-based meetings monitoring the daily work of researchers and field supervisors visiting the homes to eliminate possible outbreaks *Aedes aegypti*. The researchers accompanied the supervisors and EDCAs, and these followed the other workers for 13 days to observe the implementation of the EBS strategy field actions [50].

After the field activities, the twenty EDCAs who participated in the training were submitted to semi-structured interviews previously scheduled and carried out individually according to each worker’s availability. The interviews aimed to answer the following points: (a) the importance and contributions of *Aedes aegypti* control training; (b) Relevant knowledge, new aspects, and applicability of the EBS approach to vector control performed by EDCAs; (c) Importance and expected results of the vector control training using the EBS approach. The statements were recorded, transcribed, and analyzed according to Bardin’s Content Analysis [60].

### 2.11. Community Awareness of Appropriate Environmental Management

During home visits in the three entomological surveys, the EDCAs guided the elimination of potential *Aedes* breeding sites. It was recommended that: (a) unused small reservoirs should be sent for recycling or appropriately disposed of by the municipal government of Fortaleza; (b) small storages in use, such as buckets, drinking troughs for animals, and refrigerator’s trays should be emptied and cleaned weekly; (c) the large tanks used daily to wash clothes and for other domestic use should receive protection customized according to the residents’ interests (ex: cover with elastic, and weekly emptying and cleaning); (d) large, little-used tanks, such as waterholes, with no coupled engine and abandoned pools, should receive beta fishes; (e) the large elevated tanks should be sealed; (f) the peridomicile, such as patios, backyards, and gardens, should be cleaned and inspected weekly to eliminate any environment conducive to the development of *Aedes* (e.g., hollow in trees or walls, potted plants, and damaged gutters); and (g) drains should be screened.

### 2.12. Empty Lots: Cleaning and Risk Analysis for Arboviruses

A severe problem for communities is the inadequate destination of solid waste. Many residents dispose of these materials in open spaces such as sidewalks, squares, and vacant lots [6]. The EDCAs identified the owners or responsible for the lots, discussed the importance of eliminating potential breeding sites in these spaces, and agreed on a new visit to assess whether the site still had an infestation risk.

### 2.13. Construction of a Booklet to Prevent Arboviruses in Pregnant Women

Constructing the booklet followed the recommendations for the production of educational materials: submission of the project to the Research Ethics Committee (REC), bibliographic survey, adaptation of the material to the target audience’s language, and peer validation [61]. After surveying the scientific production, the researchers decided to involve health professionals, pregnant women, and their families to collect more information about knowledge gaps. ZIKV and CHIKV infections were recent in Latin America at the time.

Initially, we visited the Primary Health Care Units (PHCU) in the intervention areas to invite professionals from the Family Health Strategy (FHS) teams, their managers, and professionals from the Family Health Support Center (FHSC) to participate in actions involving pregnant women and partners. The two UAPS deployed in the territory were PHCU Viviane Benevides (composed of three FHS teams) and PHCU Maciel de Brito (composed of four FHS teams and one from the FHSC).

Researchers, professionals from management, FHS, and FHSC assistance met in each PHCU in the territory to discuss and plan the recruitment of pregnant women and partners to participate in the meetings, and meetings to verify and expand the previous knowledge of pregnant women and partners about preventing arboviruses during pregnancy.

The CHW first invited pregnant women at their homes and scheduled prenatal care visits. The researchers visited the PHCU to invite, dialogue about the proposed meeting, deliver a printed invitation, and record identification data and telephone contacts (to reinforce the invitation the day before the meeting). Service professionals collaborated by promoting initial contact between researchers and pregnant women stressing the importance of meetings for their health.

Three meetings with conversation wheels were held in each PHCU, lasting 90 to 120 min each. In the first meeting, a questionnaire [45] with variables on the participants’ sociodemographic profile, telephone contact, social network, and name of the CHW who accompanied them was applied. The conversation wheels aimed to promote dialogical meetings, providing new meanings to knowledge and experiences in *Aedes aegypti* control and the prevention of arboviruses. The demands experienced by pregnant women and partners were explored. They promoted the socialization of experiences, the production of reflections, and transformed participants’ ways of acting and thinking [62,63].

Thematic axes guided each conversation wheel. A better approximation of the themes was enabled with an exhibition of videos, storytelling, and images of *Aedes aegypti* breeding sites [45]. The first, second, and third themes were, respectively: “Knowledge of pregnant women and their partners on the diseases transmitted by *Aedes aegypti*”; “Vector control practices” and “Prevention of mosquito-borne diseases—care practices with pregnancy and the fetus”. All participants’ concerns were addressed at the end of each meeting. The meetings were recorded and statements were transcribed.

The recommendations on how to prepare health care materials for lowly educated people [64] were followed to construct the booklet. The criteria for the elaboration of printed educational materials were observed to make them understandable, legible, and culturally relevant [65]. The booklet’s roadmap continued with sessions that corresponded to the themes of the conversation wheels’ meetings. A graphic designer produced illustrations and layout. The Adobe Illustrator CS3 program was used to draw and color the figures and the Adobe InDesign CS6 for the layout.

Then, the validation model’s expert classification criteria [66] were used to establish peers’ eligibility. The Lattes Platform of the National Council for Scientific and Technological Development (CNPq) and the Snowball technique [67] were used for recruitment. The invitation letters were sent by e-mail along with the ICF and the electronic access link to validate the booklet, created from Google forms. Adjustments were made following peers’ suggestions, adding the most relevant ones, followed by text revision and printing.

All this construction resulted in the production of a reliable tool that can be used by professionals, pregnant women, and relatives to prevent arboviruses in pregnant women.

### 2.14. Participatory Surveillance in School Settings

First, the researchers scheduled meetings with the municipal education institutions’ managers to present the proposed research/intervention and obtain their consent and support for developing *Aedes aegypti* control actions in school settings. Then, schools in the intervention areas were visited to invite and sensitize principals, teachers, parents, students, and social mobilizers.

The researchers adapted to each school’s dynamics and built with the various stakeholders the recreational, economic, and sustainable activities for long-term continuity. The strategies employed are described below.

#### 2.14.1. Junior Brigade

The term “Brigade” is recognized in the field of public health as a work action developed by health surveillance and endemics control to eliminate possible *Aedes aegypti* breeding grounds [41]. In this research, the Junior Brigade aimed to develop a routine of inspection, treatment, or elimination of breeding sites in schools.

Four groups were established, one for each grade of the last Elementary School grades. Each group comprised eight students, an EDCA, a researcher, and a social mobilizer. Teachers were invited to participate, but they claimed they could not make time for this activity.

All sixth to ninth-graders were invited to participate. Those interested were registered in a numbered list and submitted to a draw. The selected students received a registration form [68] and a consent form to be signed by the parents. A new draw replaced those who did not present parental authorization.

After the team was set, training was promoted to expand students’ knowledge about *Aedes* control. The first training took place in a single four-hour meeting. However, the researchers realized that it would be necessary to expand the workload to maintain optimal performance. Thus, the schedule was restructured to accommodate one four-hour and two 60-min meetings. The activities were developed during student’s break time.

Psychodrama was the theoretical framework underpinning the planning and elaboration of the training program. The classic psychodrama stages are non-specific warm-up, specific warm-up, role play, and sharing, whereas the main techniques are role play and role-playing games [69]. These tools effectively build a participatory, multidisciplinary teaching-learning context that dialogues with the principles of the EBS approach; namely, knowledge for action, sustainability, and social participation [70].

The first meeting aimed to create the group identity, probe the participants’ prior knowledge, and expose *Aedes aegypti* biology and control theoretically and dialogically. This event had the following sequence: (1) Body stretching to warm up the body, release tensions and stress from the routine and prepare for the next activity; (2) Presentation of the game entitled “building the badge”. In this game, students produced their badge and introduced themselves to the group. Students were encouraged to talk about their experiences with arboviruses, their hardships, curiosities, and expectations regarding the Junior Brigade during the presentation. This information characterized the components of the social context of those involved and underpinned the learning process [71]; (3) Formation of small groups and role play. Each school year’s participants created a group. Each group planned and presented a role play about their life experiences in the context of arboviruses (dengue, Zika, and chikungunya); (4) Collective construction of knowledge and integration of the learning process with life experience. This was the moment to express how each student experienced role play, and what they observed, felt, and learned; (5) Dialogued theoretical presentation on the biological aspects involving *Aedes aegypti* and the new vector control strategies. The slides and the four phases of the vector cycle (egg, larva, pupa, and adult) were shown to illustrate the statements and consolidate knowledge; (6) Closing of the meeting with the EDCA report on his experience with the routine of visiting and inspecting *Aedes aegypti* breeding sites and presenting his work tools.

The second meeting aimed to present the EBS strategy to the students, using an adaptation of the “Knowledge Path” [72]. The researchers produced cards with the content to be discussed and randomly delivered them to some participants. Then, they started to discuss them, and students were instructed to make suggestions linking their previous knowledge with the information contained in the cards. As the themes were explored, the corresponding card was placed in the middle of the circle of students. When the last card was placed, students were invited to observe the established scheme.

The third meeting aimed to strengthen the principle of the EBS strategy: social participation. The students were encouraged to reflect on the principle of social participation and its contribution to health promotion [70] through the “The (string) ball game”. This technique initially consisted of setting the group in a circle, tossing the string to a participant who was prompted to answer the question: What do you mean by social participation? After sharing their knowledge, students twirled the string around their finger and threw the ball at another, asking the question again. After everyone’s response, the group was encouraged to observe the web formed with everyone’s involvement, connecting with the principle of social participation of the EBS strategy for the *Aedes aegypti* control. After this sharing, the web was undone in reverse, from the last to the first participant answering the question: How do I participate with my home, my neighborhood, my school with actions to control the *Aedes aegypti* mosquito?

The first, second, and third meetings were held in the classrooms. The “Junior Brigade” activity started the week following the first meeting. The students conducted a collective search for possible *Aedes aegypti* breeding sites accompanied by the EDCA, a researcher, and a social mobilizer. During the inspection, potential breeding sites were controlled, and suspicious containers were eliminated or managed, considering their specificities.

A meeting focusing on further student training was held at the end of each four-week cycle. Teams were gathered to build new knowledge, exchange experiences, strengthen bonds, and fraternize through a collective snack.

During the school holidays, the EDCA and researchers carried out the Junior Brigade until students and mobilizers resumed their activity.

At the end of four cycles (four months), school managers were consulted on the process sustainability, and they opted to continue because of the perceived gains. Each school elected a responsible adult to accompany the students in the vector control action, considering interest, availability, and involvement with the activity. Thus, the schools assumed the collective commitment to continue the Junior Brigade’s actions in partnership with the EDCA.

#### 2.14.2. Photovoice

Photovoice is a photographic representation technique used to empower excluded community members to identify, represent, and reinforce existing resources [73]. Photovoice’s application aimed to enhance students’ role in coping with *Aedes aegypti* in school settings and the community.

This technique was used with fourth to ninth-graders in two schools located in the intervention territories. Initially, the mosquito’s life cycle was displayed in the schoolyard, with models representing houses protected from *Aedes* and distributing educational materials to all attending students. Students were then invited to participate in a short course to reflect on participation and shared social responsibility in vector control actions, with a workload of eight hours [74].

A semi-structured questionnaire was applied before and after the short course [75]. On another occasion, students toured the school, with cameras and cell phones recording images related to what was discussed in the short course. Then, they participated in a focus group to expose the ideas and meanings attributed to the photos. The reports were recorded and transcribed. The researchers printed some photos and statements and promoted school exhibitions [75].

#### 2.14.3. *Aedes* Hunters

The *Aedes* Hunters action was performed with elementary schoolchildren (third to fifth grade) at the João Nunes Pinheiro municipal school. The school had two third-grade classes (third-grade A and third-grade B; one fourth-grade class, and two fifth-grade classes (fifth-grade A and fifth-grade B)). All third- to fifth-graders were invited to participate in the research. Their participation was allowed when parents signed the Informed Consent Form (ICF). The action was initiated with a training whose theme consisted of the *Aedes aegypti* Life Cycle, main breeding sites, and elimination methods.

A total of 110 schoolchildren aged 6–10 years participated in the action. All students received a preliminary home visit carried out by EDCA to map the breeding sites in the homes. Some visits were not carried out due to refusals by some parents of students. After training, students received an illustrative form (Appendix A) to search for breeding sites in their homes. The forms were collected weekly by the EDCA, and a new one was delivered so that the student’s home inspection could be monitored. The students who marked non-eliminated breeding sites or showed difficulty eliminating a specific breeding site received a visit from the EDCA to eliminate or manage the breeding site, avoiding the use of larvicide. The activity lasted three months (May, June, and August 2019). Educational activities were carried out in the school space with all students, the social mobilizer, and researchers during the action. Such actions included samples of *Aedes aegypti* life cycle, theaters with puppets, and explanations about the mosquito and control practices, ways of preventing illness from arboviruses, and home and school care.

### 2.15. Participatory Case Surveillance

The use of information and communication technologies accelerates the speed of information so that health systems can take timely actions to prevent outbreaks [76]. In Mexico, mobile telephony to pass on entomological data collected by field professionals was more agile than paper recording [77]. In Pakistan, for example, mobile phone data and climate information can predict the geographic distribution of Dengue epidemics [78]. Studies point out the relevance of integrating active surveillance and traditional surveillance to avoid underreporting cases, improve understanding of disease dynamics, and implement public health actions [79]. The use of these technologies and community participation strengthens the surveillance and prevention of these diseases [53,76].

Given the range of technology-based strategies, we aimed to build knowledge and establish participatory surveillance networks for dengue, chikungunya, and Zika cases using social networks. The process consisted of five steps:Step 1: Meetings were held with the EDCAs and FHS teams to map the various groups in the intervention areas.Step 2: Researchers visited the groups and scheduled a time to present the proposal for participatory surveillance.Step 3: The proposal was presented, and meetings were scheduled for training on topics involving arbovirus surveillance.Step 4: Four meetings to strengthen knowledge on the following topics were held: 1—group identity; 2—*Aedes aegypti* life cycle and transmission of arboviruses; 3—Case definition for arboviruses dengue, Zika, and chikungunya and 4—Social determinants and epidemiology of the neighborhood.Step 5: The WhatsApp application was employed to report suspected cases of dengue, Zika, and chikungunya.

Questionnaires with multiple-choice questions were applied before and after the workshops to gauge participant evolution regarding knowledge about the topics covered (Appendix A). Descriptive analysis was performed (absolute and relative frequencies). The percentage of correct responses was calculated in the pre and post-test. The intervention was effective when the participant had a 30% increase in the percentage of correct answers in the post-test. Fridge magnets with information on dengue, Zika, and chikungunya signs and symptoms were also distributed.

WhatsApp-reported suspected cases were investigated and conducted by municipal health professionals. Thus, no outbreaks of arboviruses in the intervention areas were reported during the period by the health notification system or by WhatsApp groups. The cases reported by community surveillance will be compared to notifications from the health system to assess the accuracy of WhatsApp notifications.

The volunteers also started to report suspected cases of COVID-19 (in 2020) and request health information from professionals through the application.

Surveillance is essential for the organization of health services, and simple strategies such as free applications are of great value for community surveillance and prevention of arboviruses’ outbreaks.

### 2.16. Comparison of Arbovirus Disease Cost to Government and Household Intervention Cost-Effectiveness

We conducted an economic analysis to evaluate the cost of arbovirosis for Fortaleza and cost-effectiveness analysis of the multisectoral intervention for the prevention and control of *Aedes aegypti*—transmitted diseases.

The municipality’s disease cost was estimated by obtaining out-of-pocket expenses from an interview conducted with patients on any arboviruses (dengue, chikungunya, Zika) reported by the surveillance system. Indirect costs were also estimated from the survey by asking arbovirosis cases older than 18 years of age about income and the number of days they were out of a job because of the disease., The number of days and the payment to caregivers was recorded (Appendix A) when the case required caregivers. The disease’s direct costs were retrieved from hospital records and included the cost of hospitalization, laboratory tests, X-rays, and other imaging tests, physician visits, drugs, and supplies. An estimated total cost of the disease was achieved with data on out-of-pocket expenses, indirect and direct costs, and the number of cases in the city [80].

Concerning the cost-effectiveness assessment, the intervention costs were estimated directly from the project’s records on expenses with supplies (nets, screens), training, materials, and informative brochures. The local Health Secretariat provided the regular program cost from the records of payments to staff, endemic workers, vector control technicians, transportation, and materials (insecticides).

Effectiveness was estimated by comparing the proportion of positive breeding sites in both intervention and control areas before and after the intervention. The Incremental Cost-Effectiveness Ratio (ICER) was calculated as the ratio of the difference in effectiveness and the difference in costs between the control and intervention area. The parameter adopted was that ICER should not exceed 1–3 per capita GNP. ICER for Brazil was 14,120, and 1–3 per capita GNP is U$ 8353–25,059.

### 2.17. Chikungunya’s Chronicity

The researchers, EDCAs, and community health workers searched for data on people with chronic chikungunya [81]. The identified people received a visit from the EDCAs and were invited to participate in an interview about their health condition after being affected by chikungunya.

A questionnaire with open-ended questions was used. The statements were recorded, transcribed, and analyzed using content analysis [60], in the light of the intersectionality framework [82] and the anthropology of pain [83].

### 2.18. Sealing Acceptance and Sustainability

During entomological visits (September–December 2018 and February–May 2019), the sustainability of water tank sealing carried out with project resources was assessed. In properties where a violation was identified, 38 were randomly selected to investigate why the screen was removed or broken.

In August 2019 the EDCAs visited the residents and asked for permission to schedule an interview between the researchers and sealing beneficiaries. Residents determined the day and time of the interviews. Some residents were not found, and others refused to participate in the interview. In the end, the sample consisted of 23 people. The interviews were recorded, transcribed, and analyzed using the content analysis technique [60].

### 2.19. Governance of Actions to Combat Arboviruses

We aimed to identify the dynamics of the actors involved in the prevention and control of vector-borne diseases in Fortaleza and their management, leadership, interaction between them, power relationships, decision-making spaces, technical capacity, and community participation mechanisms. State and municipal managers were invited by telephone to participate in the research. After their acceptance, a semi-structured questionnaire (Appendix A) and ICFs were forwarded to their e-mails. Data were organized into categories, using content analysis [60].

Municipal vector control actions occurred in the four study areas throughout the intervention period. The following activities were carried out: the *Aedes aegypti* Infestation Index Rapid Survey (AaIIRS); cleaning efforts; insecticide applications with a vehicle-fitted nebulizer; the ultra-low volume (ULV); home visits to inspect water storage tanks or containers; intensification of home visits and cleaning efforts, when AaIIRS identified strata at risk of an outbreak (building infestation index > 3.9%); and information and education activities carried out by social mobilizers.

The linkage between researchers and managers strengthened vector control actions, mainly at the municipality level. The researchers observed that managers were compelled to also implement more actions in the control areas, that is, cleaning efforts or intensifying home visits because of actions developed in the intervention areas. Several endemic workers asked the researchers to expand the EBS strategy actions to the control areas because they believed they were effective.

### 2.20. Data Management

The data collected in the entomological and cost of arboviruses to households’ assessments were recorded by field workers on forms, processed in the Epi Info 7 software and analyzed in the STATA^®^ software version 15. Secondary data on the costs of arboviruses to the government were recorded and analyzed in Microsoft Excel software, and quantitative data were collected in schools and the community.

The audios from the interviews focus groups and conversation circles were recorded with a Sony digital recorder, transcribed, and categorized according to Bardin [60]. Data on the training of health workers, Junior Brigade, *Aedes* Hunters, community case surveillance, and governance were manually categorized. Photovoice data were analyzed using IRAMUTEQ, a free and open-source software, while chikungunya’s chronicity data were analyzed with the NVIVO^®^ software. Each researcher was responsible for selecting the proper software used in the analyses and processing their data.

## 3. Data Analyses

### 3.1. Outcomes

#### 3.1.1. Primary Outcomes

Initially, the frequency of small storage tanks was analyzed. Then, the proportions of households with each small storage tank type were calculated, in the two areas (Intervention and Control), during the three entomological surveys (ES Before, During, and After). Then, the proportions of the Intervention Area of two consecutive ES (ES Before versus ES During; ES During versus ES After; ES Before versus ES After) were compared using the McNemar Chi-square test. In the same way, the proportions of the Control Area of two consecutive ES (ES Before versus ES During; ES During versus ES After; ES Before versus ES After) were compared.

The same procedure was adopted to compare: (i) the proportions of households with each of the different types of small infested storage tanks; (ii) the proportions of households with each of the several types of large storage tanks; (iii) the proportions of households with each of the several types of large infested storage tanks.

The proportion of infested households or buildings (House Index, HI) of the Intervention Area and Control Area, for each of the three Entomological Surveys (ES Before, ES During, and ES After), were compared using Pearson’s Chi-square test. In turn, the proportion of Container Index (CI) of the Intervention Area and Control Area, of each of the three Entomological Surveys (ES Before, ES During, and ES After), were compared using the Student *t*-test. Likewise, the Number of positive containers per 100 houses inspected (Breteau Index, BI) the Intervention Area and Control Area of each of the three Entomological Surveys (ES Before, ES During, and ES After) were compared using Poisson regression.

The proportion of Small Infested Storage Tanks (the sum of all types) in the Intervention Area and Control Area, of each of the three Entomological Surveys (ES Before, ES During, and ES After), were compared using Pearson’s Chi-square Test. A similar procedure was adopted to compare the proportion of infested Large Infested Storage Tanks (the sum of all types), from the two areas, in each entomological survey.

The temporal evolution of the HI, CI, and the BI, over the three entomological surveys, in the two areas (Intervention and Control), were analyzed through random intercept regression, including an interaction term for the variables Area and Entomological Survey. This interaction term allows the comparison of the slope of two lines (one for each area) at two different times, which correspond to the infestation of the two areas during two successive entomological surveys. A logistic regression model was used for the HI, and linear regression models were used for CI and BI.

The temporal evolution of the Number of Small Storage Tanks (NSST) over the three entomological surveys in both areas was analyzed using random intercept regression; specifically, using Poisson Regression models, including an interaction term for variables Area and Entomological Survey. The interaction term allows the comparison of the slope of two lines, at two different times, which correspond to the number of deposits in the two areas (Intervention and control) during two successive entomological surveys.

The temporal evolution (over the three entomological surveys) of HI, CI, BI and NSST of the prediction of the respective regression equations, plus the p-value of the interaction term (Area X Survey) were presented in graphs.

#### 3.1.2. Secondary Outcomes

The secondary outcomes captured from focus groups and in-depth individual interviews included: (1) acceptability and sustainability of the sealing of large elevated water reservoirs; (2) student knowledge about transmission and prevention of arboviruses; (3) establishment and sustainability of groups of students to monitor and eliminate potential *Aedes* breeding sites at their institutions; (4) establishment and sustainability of a community surveillance network for arbovirus cases; (5) Intersectionality in women with chronic chikungunya; (6) *Aedes* control governance; and (7) EBS strategy cost-effectiveness and direct and indirect cost of arboviruses [84]. The results were analyzed at the household or individual level, except for governance and the costs of arboviruses to the government for Fortaleza.

### 3.2. Ethics and Dissemination

The Ethics and Research Committee of the State University of Ceará approved the research. The study design, ethics, provisional analyses, and risks to which participants were exposed were assessed. The risk of embarrassment was observed in the interviews, conversation wheels, culture circles, and training, which were minimized by the researchers’ mediations and the setting’s preparation to prevent external interference. Any risk of damaging the roof of homes whose elevated tanks were sealed was identified and minimized by selecting specialized and experienced people to carry out this action.

Adult participants formalized their agreement by signing the consent form; minors signed an assent form, and their parents or guardians, the ICF authorizing their participation.

Initial analyses indicate that the EBS strategy is effective on a large scale. These results are available on the IDRC platform (https://idl-bnc-idrc.dspacedirect.org/bitstream/handle/10625/59057/IDL%20-%2059057.pdf?sequence=2&isAllowed=y), and in other scientific publications [45,50,68,74,75]. The remaining results will be published in peer-reviewed journals and on the website of the State University of Ceará. Data requests can be sent to researchers at the State University of Ceará after the publication of the unpublished results.

## 4. Discussion

This large cluster, non-randomized, controlled trial on the EBS strategy effectiveness in highly socially vulnerable territories with an epidemiological risk for arboviruses provided the community with training to act on the mosquito’s vector control in their homes and schools, and establish a community surveillance network of arbovirus cases.

The technologies produced offer subsidies and guidelines for replicating vector control actions, environmental management, training of professionals, students, and community groups, implementing surveillance networks in schools and the community, and mobilizing pregnant women and professionals to prevent arboviruses during pregnancy.

The lack of randomization and blinding of field workers and evaluators are limitations of the study design.

Arbovirus prevention strategies should be developed with the community’s participation because people inhabiting that territory are the most knowledgeable about the weaknesses and potentialities permeating the health context in question. Studies point to the effective social participation in vector control and the benefits related to lower economic costs, as these are simple activities (e.g., elimination of potential breeding sites) and do not require qualified labor [24,33,37].

This study evidenced that people had little time to worry about vector control because they were working to ensure immediate livelihood. Also, the care of elevated tanks is neglected and delegated to professionals who repair broken water pumping gear parts. Many people did not follow a routine for cleaning and sealing the tanks, resulting in waste accumulation and infestation.

Preliminary data confirm that the sealing of large tanks reduced the infestation. However, this was not followed in some properties because those responsible did not pay attention to the correct method of fixing of the screens when proceeding with repairs.

As a palliative measure, we recommend training for professionals working in the maintenance of large elevated tanks. A more lasting effective measure would be regular water supply. Thus, large tanks would not be necessary to ensure access to water. Stakeholder participation is also essential in the whole process, as these are people affected by the problem or influential in decision-making and project implementation [85]. Since arboviruses are problems encompassing several societal sectors, one must expand the perspective and identify the largest number of people interested in a transdisciplinary [86] and governance perspective [87]. The set of knowledge gathered or produced should point to solutions applicable by managers. Thus, proposals should be clear and feasible.

Therefore, we emphasize the relevance of this protocol as a governance tool for fighting the mosquito, which provides us with several tools to assess the feasibility, impact, sustainability, and acceptability of a large-scale EBS strategy for reducing *Aedes aegypti* density. The products are lightweight technologies, such as forms, questionnaires, flowcharts, training schedules, and informative materials that guide actions based on syntheses built by the authors involved.

Gathering community, government, professionals, and scholars to combat *Aedes aegypti* is essential to reduce the social and economic impact of infections such as dengue, chikungunya, and Zika. As a result, we recommend implementing the EBS strategy on a large scale, mainly in highly socially vulnerable households with a high epidemiological risk.

## Figures and Tables

**Figure 1 ijerph-18-01278-f001:**
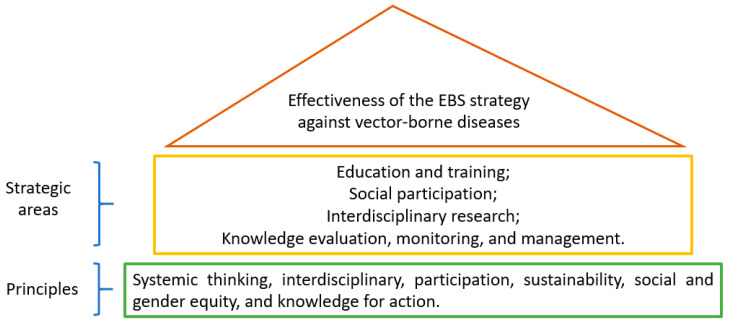
Principles and strategic areas of action of the Eco-Bio-Social (EBS) strategy.

**Figure 2 ijerph-18-01278-f002:**
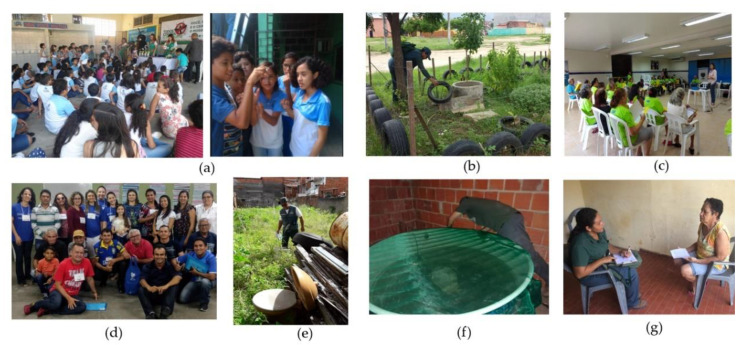
Intervention. (**a**) school activities, (**b**) elimination of breeding sites, (**c**) community groups, (**d**) training in EBS strategy for endemic diseases’ workers, social mobilizers, AVISA, and stakeholders, (**e**) cleaning of vacant lots, (**f**) sealing of water tanks, (**g**) study on the cost of arboviruses to the community.

**Figure 3 ijerph-18-01278-f003:**
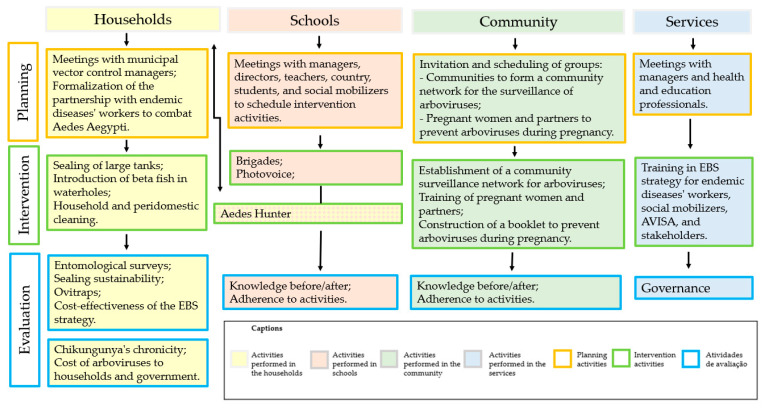
Flowchart of the developed activities.

**Figure 4 ijerph-18-01278-f004:**
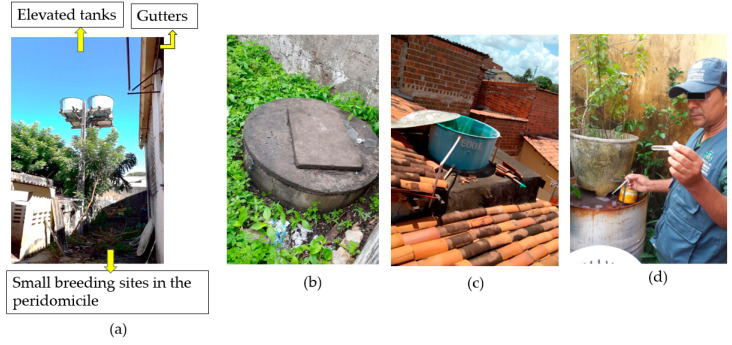
Breeding sites. (**a**) Elevated tanks, gutters and small breeding sites, (**b**) waterhole, (**c**) elevated tanks, (**d**) flower pot and large drum.

**Table 1 ijerph-18-01278-t001:** Sample size of groups that participated in the intervention.

Groups	Sample Size
EBS Training Course for professionals and interested parties	38
Pregnant women and partners	38/03
School brigade	136
Photovoice	55
*Aedes* hunters	110
Community surveillance	60

**Table 2 ijerph-18-01278-t002:** Summary of the research activities schedule.

Activities	2017		2018									2019					2020	
	JJASON	D	JF	M	AM	J	J	A	S	ON	D	J	FMAM	JJA	SON	D	J	FMA
Feasibility study																		
Baseline * or ES before																		
EBS training																		
Sealing of water tanks																		
Booklet elaboration																		
School surveillance																		
Community surveillance																		
Environmental management																		
ES during																		
ES after																		
Sealing sustainability																		
chikungunya chronicity																		
Cost-effectiveness study																		
Governance																		

* The implementation of the EBS strategy started immediately after the construction of the baseline. The background color marks the period in which the activities were carried out (baseline: JFMAM 2018).

## Data Availability

The data presented in this study are available on request from the corresponding author. The data are not publicly available due to uniqueness.

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
