# Peer review of "Scaling up of Eco-Bio-Social Strategy to Control Aedes aegypti in Highly Vulnerable Areas in Fortaleza, Brazil: A Cluster, Non-Randomized Controlled Trial Protocol"

_ijerph, 2021, doi:10.3390/ijerph18031278_

Round 1

Reviewer 1 Report

The present study brings a fascinating and complex protocol that was implemented in Fortaleza to control Aedes aegypti. Undoubtedly, this study will be highly used as a reference for similar interventions and it is outstanding in terms of the main approaches and their importance. It has high merit for publication, and I am delighted to learn about such a very difficult and profound action program that took place on a very real ground of arboviruses’ outbreaks. I consider this manuscript accept after a minor review, which needs to encompass the major aspects described below, together with the minor points addressed afterward.

Major modifications

I have 3 main points that will improve the manuscript in my view, and facilitate readability as a protocol:

1-The authors must include a Figure with a nice and well-detailed flowchart with all the activities organized hierarchically, and their relationships. If the protocol is presented with this organized overview, it will be easier for readers to organize in their heads all the process to accomplish the Eco-Bio-Social strategy. If possible, the sections would follow the main flowchart. It will be an important figure that will enhance the citations and application of this protocol.

2- The description is detailed for every single section. However, the protocol is too long. A better organization will help to navigate it, however, the authors should exercise to eliminate unnecessary/redundant information. More images (figures)  can help to deal with this issue as well.

3-  The entire protocol can be enriched by images from this experience in the field. It is a so rich and unique one, and unfortunately, this version is not contemplating several aspects of the study with images – For instance: How are these local breeding sites? Environmental aspects of the communities? Neighborhoods? Ovitrap Setup? Pamphlets? School activities?, etc… Of course, respecting the ethical aspects.

4- A major critique. The authors are proposing a protocol to be published before the experimental data. “A manuscript with the results of the primary study will be published in a peer-reviewed journal. 759 Different manuscripts will be written for each of the secondary objectives, and these will also be 760 submitted for publication in peer-reviewed journals. 761

Preliminary data were disseminated at scientific congresses and to the managers to keep up to 762 date on the entomological indices of the households. Eight postgraduate studies will be published, 763 three of which are theses and five dissertations, on the website of SUC’s Graduate Program in 764 Collective Health. The final technical report is available on the IDRC platform [81]. 765

Requests for data can be sent to researchers at the State University of Ceará after the study and 766 the publication of the main manuscript. 767”

Are the authors considering the first study already published as a proof of principle for this protocol? How can I use this protocol if I do not know whether it will work or not in terms of effectiveness to control Aedes aegypti population? How do the authors deal with this concern? Is this a concerning point? I would like to hear from the authors.

Minor modification:

Introduction:

Line 57: It could be more precise. It is not attracted by human blood…. Human odors, for instance

Line 64 and 65: A brief explanation on why it is not possible would be interesting: Are the authors mentioning all the aspects here – inexistence of the vaccines, poor vaccination program, community willing to be vaccinated, all together? Better to clarify this point, since the authors just mentioned about the lack of conditions in these communities.

Methods:

Line 152: Please, replace ZIKAV by ZIKV. Line 439 as well and possible other occurrences.

Line 155-156. The sentence can be modified to include “Table I” : “The number of participants for each intervention activity is shown in Table 1”. Below, please, replace Figure 1 with Table I (if this is the journal rule for the table number representation – or Table 1) and improve the formatting of Table I. It can be better aligned and having its own title. It is not necessary to repeat “pregnant women and partners twice -   Pregnant women/partners                             38/3

Or

Pregnant women                                       38

Partners of pregnant women                 3

Lines 190-192 – How were the costs collected/ examined?

Lines 195 to 196: Modify as suggested above to Table II. Please, cite Table II in the text before it.

Line 248- “in suburban neighborhoods [6].Due to water”. Please, add a space after the period.

Line 293: “A before-and-after test instrument” – Are these available as supplements for this study protocol? How did the authors evaluate the effectiveness of the training quantitatively?

Lines 328-331 – This paragraph is extremely contradictory and misleading for the entire study. I am not sure whether the meaning is clear to me? Are the authors mentioning that Ae. aegypti does not breed at home (domestic breeding sites)? Is it a relation with the sizes of the breeding sites? My interpretation was that positive and productive breeding sites were left unattended, is this well interpreted? Is this correct? Ethical? Please, clarify this statement. I am not sure if “waste time” is the appropriate term here. A lot of problems in this sentence. If the point was to evaluate the status of infestation, not to eliminate it after detection, it needs to be explained in another way.

Line 337 – What were the species keys used for it? Citations?

Reviewer 2 Report

A very interesting and useful report. Just some technical remarks:

It is worth mentioning in the abstract that Aedes aegypti is a primary vector of all three viral pathogens in the world and in Brazil, particularly;

sometimes you write Dengue with a small letter (lines 49, 108, 151, maybe more);

where are Appendix A and B?

Best wishes.

Reviewer 3 Report

This manuscript is presented as a protocol and describes a complex and ambitious design for enhanced prevention and control of arboviruses in Fortaleza, Brazil. The public health impact of Aedes vectored arboviruses in Brazil is overwhelming, so a complex assessment of comprehensive control and prevention is of extreme value. The target strategies are numerous, including education of various population segments, training and mobilization of citizen scientists for surveillance and control, standard entomological surveys over time, long-term control solutions, and cost comparison between intervention and case management estimates.

While the overall writing is a little clunky with sometimes wordy sentence structure, it is mostly grammatically sound. I have highlighted a few instances where readability could be improved, but an additional focused read-through by the authors should be performed. Throughout the manuscript there is a preponderance of unwarranted capitalizations. Please verify that journal guidelines for capitalization of proper nouns are being followed. For instance, typically dengue and chikungunya viruses are not capitalized, but Zika virus is because it refers to the location, Zika forest. 

I do have a major concerns regarding the limited detail regarding how effectiveness for each strategy will be assessed. The authors state the general research question as (lines 107-110) “how effective is the EBS strategy on a large scale compared to routine vector control activities…”. However, in the design description, I do not see a control site where just routine vector control activity is occurring that would allow the comparison between the two methods. Primary entomological outcomes appear only to be comparisons between the before, during, and after periods of the EBS method of intervention. Those are valuable comparisons, but do not address the question of whether EBS methods result in increased control/prevention as compared to the standard, routine vector control methods. Either a control “standard intervention” field site needs to be included or the general research question needs to be adjusted to fit what conclusions can actually be drawn based on the experimental comparisons being made.

One of the biggest challenges of any intervention strategy revolving around education and citizen/resident mobilization is the lack of lasting compliance and therefore drop in control/prevention efficacy over time. Does this intervention include follow up assessments at monthly or even 1 year post completion to see if behavior changes are maintained? Any insight into what sort of variables might be important to improve long-term behavior changes would go a long way to helping with control.

Minor comments

Introduction

General – correct capitalization of dengue, chikungunya, and Zika virus throughout the manuscript as well as improperly capitalized general names throughout the manuscript

Line 66: remove the “and” in “environment, and those mentioned above…”

Line 67: remove the “and” in “mentioned above, and high…”

Line 77: Please add a sentence or two adding a general description of the Eco-Bio-Social strategy for arbovirus control

Materials and methods:

Line 152: Abbreviations of the viruses should have been defined at the very first mention of them in the introduction.

Line 159: how were households selected?

Figure 2: It appears in the timing that the intervention measures for sealing water tanks, booklet elaboration, school surveillance, and environmental management were all occurring during the baseline measurements. That would bias and likely minimize any effect size you might see from early interventions. Is there a reason why these overlapped? It would be valuable to include the entomological survey periods (before, during, and after) on this table as well.

Line 270: Do the authors mean “eliminate potential Aedes breeding grounds around the homes” as opposed to what is written, “…grounds in the homes”

Line 286-288: The last sentence of this paragraph is difficult to follow and should be rephrased

Section 2.8 – the authors state that entomological surveys were not done at all locations. I think it would be valuable to describe how the sights where surveys were done were chosen so as to rule out any potential selection bias of the sites.

Line 328-329: “We observed that the number of Aedes aegypti larvae and pupae in domestic breeding sites is minimal in Fortelaza”. Please elaborate on that. Are they using alternative sites (tree holes, more urban settings, etc) or just present in low populations? Also, “sub-adult developmental sites” is more accurate than “breeding site”.

Line 387: Suggest placing a comma between “teachings and Freire”

Line 392: Suggest changing “theoretical meetings” to “theory-based meetings” throughout this section

Line 395: If it hasn’t been already, please define “FWAT”

Line 404: Suggest removing “occurred in real workspaces, that is,” from the sentence.

Line 417: Remove the comma after “visits” and add a comma after “surveys”

Line 454: remove the “and” after “women,”

Line 507: A four-hour meeting seems very long for keeping elementary school kids attentive. Will this happen during school hours or outside of them?

Junior Brigade section: It is unclear from the description if the pre-training for the junior brigade actually included hands on field experience or just slides. Please clarify.

Lines 623-628: Please clarify this section. No outbreaks of arboviruses in the intervention areas were reported – was that through the WhatsApp reports? Will this be correlated to public health case reports to assess accuracy?

Line 629: Suggest changing 2.16 header to “Costs of arbovirus disease cost to government and household intervention cost-effectiveness”

Section 2.16: The authors use the proportion of positive breeding sites as the measure of effectiveness; however, the comparison is between cost of intervention and medical cost. If the goal is to reduce medical costs, should the measure of effectiveness be a reduced number of cases in the treatment arm?

Section 2.18 Sealing acceptance and sustainability – please clarify how long after the initial sealing was the follow up assessment done

Section 3 outcomes: This study has a lot of arms. It would be valuable to include specific measurements/comparisons for each outcome in this section. For instance, line 701 states “small storage tanks were analyzed”. Were they analyzed for the number of openings, the size of opens, the presence of sub-adults, the density of infestation? Please clarify.

Line 705: what is meant by “proportions of the control area”?

Please replace “proportions” with proportion throughout the section

Line 742: change “knowledge of students” to “student knowledge”

Line 754: change “A risk” to “Any risk”

Line 755: change “identified, which was minimized” to “identified and minimized”

Line 757-758: Please clarify this sentence regarding minor assent forms

Round 2

Reviewer 3 Report

The authors have done a comprehensive job of correcting and clarifying methods and designs where necessary. I have a few minor comments remaining. 

  1. The authors seem to still have inconsistency with capitalization of dengue and chikungunya. I recommend using the author guidelines to follow the journal's requirements. If those aren't helpful, I suggest the ICTV website https://talk.ictvonline.org/information/w/faq/386/how-to-write-virus-species-and-other-taxa-names. Similarly, I suggest using the "find and replace" feature in your word processor so you don't have to find and change them all individually. 
  2. Please double check the edited sentence in lines 66 and 67.I think you want to change "para Zika e Chikungunya" to "for Zika and chikungunya". In line 67 the authors state the dengue vaccine is only safe for "seropositive". Do you mean "non-seropositive individuals"?
  3. In figure one in the principles section, please change "transdisciplinarity" to "interdisciplinary" or another more apt term.
  4. Line 119 suggest inserting "solely" before the second "vector control routine" might help clarify the sentence
  5. Line 424: remove the comma after "teachings"
  6. Line 673 - the heading for 2.16 - I incorrectly suggested using "Costs of arbovirus disease cost..." when I meant to suggest "Comparison of arbovirus disease cost..." please correct my mistake. 
  7. Line 855-856: suggest adding "method of" between "correct" and "fixing"
